

# Higher carbon sequestration potential and stability for deep soil compared to surface soil regardless of nitrogen addition in a subtropical forest

Chang Liao[1,2], Dong Li[1,2,3], Lin Huang[1,2], Pengyun Yue[1,2], Feng Liu[1] and Qiuxiang Tian[1]

[1] Key Laboratory of Aquatic Botany and Watershed Ecology, Wuhan Botanical Garden, Chinese Academy of Sciences, Wuhan, China
[2] University of Chinese Academy of Sciences, Beijing, China
[3] College of science, Tibet University, Lasa, China

Corresponding author
Qiuxiang Tian,
tianqiuxiang@wbgcas.cn

## ABSTRACT

**Background**. Labile carbon input could stimulate soil organic carbon (SOC) mineralization through priming effect, resulting in soil carbon (C) loss. Meanwhile, labile C could also be transformed by microorganisms in soil as the processes of new C sequestration and stabilization. Previous studies showed the magnitude of priming effect could be affected by soil depth and nitrogen (N). However, it remains unknown how the soil depth and N availability affect the amount and stability of the new sequestrated C, which complicates the prediction of C dynamics.

**Methods**. A 20-day incubation experiment was conducted by adding $^{13}$C labeled glucose and $NH_4NO_3$ to study the effects of soil depth and nitrogen addition on the net C sequestration. SOC was fractioned into seven fractions and grouped into three functional C pools to assess the stabilization of the new sequestrated C.

**Results**. Our results showed that glucose addition caused positive priming in both soil depths, and N addition significantly reduced the priming effect. After 20 days of incubation, deep soil had a higher C sequestration potential (48% glucose-C) than surface soil (43% glucose-C). The C sequestration potential was not affected by N addition in both soil depths. Positive net C sequestration was observed with higher amount of retained glucose-C than that of stimulated mineralized SOC for both soil depths. The distribution of new sequestrated C in the seven fractions was significantly affected by soil depth, but not N addition. Compared to deep soil, the new C in surface soil was more distributed in the non-protected C pool (including water extracted organic C, light fraction and sand fraction) and less distributed in the clay fraction. These results suggested that the new C in deep soil was more stable than that in surface soil. Compared to the native SOC for both soil depths, the new sequestrated C was more distributed in non-protected C pool and less distributed in biochemically protected C pool (non-hydrolyzable silt and clay fractions). The higher carbon sequestration potential and stability in deep soil suggested that deep soil has a greater role on C sequestration in forest ecosystems.

# INTRODUCTION

Soil organic carbon (SOC) is a major carbon reservoir in the terrestrial ecosystems and its slight change would have a considerable impact on the global carbon (C) balance (*Cotrufo et al., 2015*). Litter and root exudation provide a large amount of labile substrate to soil microorganisms (*Paul, 2016*), mediating C cycling in the terrestrial ecosystems (*Cheng et al., 2014*; *Paul, 2016*). First, inputs of labile C can greatly enhance native SOC mineralization, which is termed as priming effect (PE) (*Blagodatskaya & Kuzyakov, 2008*; *Kuzyakov, 2010*; *Blagodatskaya et al., 2007*). Second, part of the added labile C can be retained in the soil to compensate the SOC loss caused by PE (*Ohm, Hamer & Marschner, 2007*; *Liang & Balser, 2012*; *Cotrufo et al., 2013*; *Cotrufo et al., 2015*; *Liang, Schimel & Jastrow, 2017*). Hence, labile C input studies should consider net C balance between the retained C and the primed C (*Griepentrog et al., 2014*). Additionally, the primed C was found to be originated from stable SOC (*Fontaine et al., 2007*; *Blagodatskaya et al., 2011*; *Derrien et al., 2014*). If the retained C is less stable than the primed C, the long-term effect of C input on SOC storage remains uncertain. Therefore, it is necessary to investigate how much exogenous C can be sequestrated in soil and the stabilization of the new sequestrated C (*Powlson et al., 2014*; *Janzen, 2015*). This new knowledge can show new insights into our accurate assessment of the impact of labile C input on soil C pool.

Current studies suggested that the new sequestrated C has faster turnover time and lower stability (*Derrien et al., 2014*; *Van Groenigen et al., 2017*). However, the reason underlying the phenomenon was poorly understood. It is necessary to reveal the SOC stabilization mechanism for the accurate evaluation of SOC stability. Soil fractionation analysis is frequently used to study the stabilization mechanisms of SOC by separating SOC into different C fractions. These SOC fractions correspond to different mean residence times and stabilization mechanisms (*Christensen, 1992*; *Christensen, 2001*; *Six et al., 2002*; *Von Lützow et al., 2007*). Therefore, studying the dynamics of labile C incorporation into these soil fractions can elucidate the new C stabilization mechanisms and evaluate its stability.

Deep soil (below 30 cm) contains more than half of the total soil C stocks (*Rumpel & Kögel-Knabner, 2011*). The response of deep soil to labile C input is thus important to terrestrial C balance. Previous studies generally found that deep soil had a stronger PE than surface soil due to its soil physio-chemical and microbial properties (*Tian et al., 2016*; *Wang et al., 2014b*). It is urgent to know how much and how long the exogenous C can be retained in soil to evaluate the long-term C sequestration in forest ecosystems.

Soil C cycling under labile C input can be mediated by nitrogen (N) availability (*Chen et al., 2014*; *Qiu et al., 2016*). Many studies mostly found that higher N availability could reduce soil PE by regulating soil microbial activity and metabolic efficiency (*Wang et al., 2014a*; *Wang et al., 2014b*; *Chen et al., 2019*). The changed microbial properties might further affect exogenous C mineralization and C sequestration. Field researches showed N addition could significantly regulate SOC balance mainly through indirect effects on vegetation C input (*Liu et al., 2019*). However, the direct impact of N addition on new C sequestration was unclear.

In order to identify the effects of soil depth and N availability on the soil C sequestration potential and the stability of the new sequestrated C, surface soil (0–10 cm) and deep soil (30–60 cm) from a subtropical forest were incubated for 20 days with the addition of $^{13}$C labeled glucose and $NH_4NO_3$. Soil $CO_2$ efflux rates and $\delta^{13}$C values were measured during the incubation. SOC was fractioned into seven fractions and collectively divided into three functional pools by a combination of density, particle and chemical methods to elucidate C stability. Previous studies showed that SOC in deep soil was further away from C saturation than that in surface soil, and deep SOC was less decomposable than surface SOC (*Stewart et al., 2008*; *Poirier et al., 2013*; *Derrien et al., 2014*; *Van Groenigen et al., 2017*). Therefore, we hypothesized that the deep soil could retain higher proportion of exogenous C than surface soil, and the new C in the deep soil was more stable. We also hypothesized that N addition could increase the new C sequestration and stability.

## MATERIAL AND METHODS

### Soil collection

Soil samples used in this experiment were collected in an evergreen and deciduous broad-leaved mixed forest located at Badagongshan National Research Reserve (29°46.04′N, 110°5.24′E) in Sangzhi county, Hunan Province. More detailed site description could be seen in *Tian et al. (2016)*. Field sampling was verbally permitted by Zhirong Gu, who is a worker at the Badagongshan National Research Reserve.

The soils were collected from two depth intervals: 0–10 and 30–60 cm (representing the surface soil and deep soil, respectively) through digging a trench. The 10–30 cm soil included mixed samples from mineral A layer, transition layer, and Bts layer. Therefore, we did not consider this layer in our study. The soils were homogenized and then sieved (two mm). The coarse roots and visible residues were picked out during the sieving. Soils were stored below 4 °C until further incubation. SOC and TN contents were 131.7 and 7.9 mg g$^{-1}$ in surface soil, and was 35.7 and 2.7 mg g$^{-1}$ in deep soil (Table 1). The clay contents were 22.6% and 39.6% in surface soil and deep soil, respectively.

### Experimental design and soil incubation

The incubation experiment included three treatments: soil without addition (Control), soil with glucose addition (Glu), and soil with combined additions of glucose and N (Glu+N) with six replicates. For each soil, the amount of added glucose-C corresponded to 100% of soil microbial biomass C (MBC) (surface soil, 2061.8 μg C g$^{-1}$ soil; deep soil, 154.7 μg C g$^{-1}$ soil). This quantity has been widely adopted in priming experiments (*Li et al., 2017*; *Chen et al., 2019*). The amount of N addition corresponded to the C: $N = 10$ of the added substrate with $NH_4NO_3$ (surface soil, 206.18 μg N g$^{-1}$; deep soil, 15.47 μg N g$^{-1}$ soil). The control received the same amount of distilled water. For incubation, about 60 g soil (equivalent to 30 g dry soil) for each replicate was placed into an individual 250 mL Erlenmeyer flask. Soil samples were pre-incubated at 20 °C for 5 days in the dark condition. After pre-incubation, each replicate of Glu and Glu+N treatments was amended with aliquots of glucose solution (uniformly labeled, $\delta^{13}$C = 2,000‰) with or without N source ($NH_4NO_3$). For the control treatment, equivalent distilled $H_2O$ (three mL) was
**Table 1  Characteristics of the soil samples.**

| Variables | Surface soil (0–10 cm) | Deep soil (30–60 cm) |
|---|---|---|
| SO (mg g$^{-1}$) | 131.7 | 35.7 |
| Total N (mg g$^{-1}$) | 7.9 | 2.7 |
| C/N | 16.7 | 13.4 |
| $\delta^{13}$C (‰) | −28.2 | −25.8 |
| LF (%) | 4.04 | 0.56 |
| Sand (%) | 3.18 | 3.98 |
| Silt (%) | 70.17 | 55.86 |
| Clay (%) | 22.61 | 39.60 |
| WEOC(%) | 0.33 | 0.33 |
| LF C (%) | 10.14 | 4.11 |
| Sand C (%) | 0.72 | 2.38 |
| H-silt C (%) | 28.17 | 33.78 |
| NH-silt C (%) | 36.58 | 17.11 |
| H-clay C (%) | 12.03 | 29.11 |
| NH-clay C (%) | 12.03 | 13.18 |
| MBC (μg C g$^{-1}$ soil) | 2061.8 | 154.7 |

**Notes.**

Soil microbial biomass C (MBC) was measured after 5 days of pre-incubation. LF%, sand%, silt% and clay% represented the mass percentage. The WEOC (%), LF C (%), sand C (%), H-silt C (%), NH-silt C (%), H-clay C (%) and NH-clay C (%) were the percentage of each fraction C in total SOC. The mass recovery after fractionation procedure was over 99%.

added to the soil samples. Then, soil samples were incubated in dark under the ambient air condition for 20 days. During the incubation, the soil water content was maintained at 65% water-holding capacity by weighing the flask every 4 days. Soil $CO_2$ efflux rates were measured on days 0, 1, 2, 3, 4, 7, 10, 15 and 20. After the 20 days incubation, the $CO_2$ efflux rate tended to be constant. Thereafter, destructive samplings were conducted. Three replicates were used to measure the MBC, and the other three replicates were oven dried at 60 °C to analyze SOC fractions.

## Measurements of $CO_2$ efflux rates

At each measurement time, three of the replicates were randomly chosen to measure $CO_2$ efflux rates by an infrared gas analyzer (IRGA; EGM-4, PP Systems, USA). Another three replicates of each treatment were chosen to determine $\delta^{13}$C of the released $CO_2$ by Carbon isotope analyzer (912-0003, LGR, USA). The detailed procedures could be seen in *Tian et al. (2016)*.

## Measurement of MBC

MBC was determined by the chloroform fumigation extraction method (*Vance, Brookes & Jenkinson, 1987*). Soil samples were divided into two 10 g fresh soil. One was fumigated with ethanol-free chloroform for 24 h followed by extraction with 0.05 mol L$^{-1}$ K$_2$SO$_4$ (shaken for 30 min) and the other was extracted immediately with 0.05 mol L$^{-1}$ K$_2$SO$_4$ (shaken for 30 min). The extraction was then determined by Total Organic Carbon Analyzer (Vario TOC, Elemental, Germany). The MBC (difference in K$_2$SO$_4$-extractable C between fumigated and non-fumigated samples) were corrected using universal conversion factors

of 0.45 (*Garcia-Pausas & Paterson, 2011*). The extractable C content for the non-fumigated soil samples was considered as dissolved organic carbon (DOC).

## SOC fractionation

Soils were fractionated using a combination of density, particle and chemical protocol, adapted and modified from *Denef et al. (2013)* and *Six et al. (2002)* (Fig. S1). Before SOC fractionation, water-extractable organic carbon (WEOC) was extracted by shaking 5 g of dried soil (<2 mm) in 20 mL of deionized water on a shaker for 2 h. After extraction, the solid residue was separated into light fraction, sand, silt and clay. Briefly, the oven-dried (60 °C) solid residue soil was placed in a 50 mL centrifuge tube and 25 mL of NaI solution with a density of 1.85 g cm$^{-3}$ were added. Tubes with the soil-NaI mixture were shaken in a shaker table at 300 rpm for 2 h. Then, the samples were centrifuged and the floating light fraction (LF) (LF < 1.85 g cm$^{-3}$) was transferred onto the microfiltration membrane and filtered under vacuum. The remaining heavy fraction was washed with deionized water to remove NaI and then sieved through a 53 $\mu$m screen to separate the sand (>53 $\mu$m) fraction from the silt and clay. Silt and clay fractions were separated through wet centrifugation (127 g for 7 min for silt, and 1,730 g for 15 min for clay, and then further hydrolyzed in 6 mol L$^{-1}$ HCl at 95 °C for 16 h. The C in the suspension and residue were considered as hydrolyzable C fraction (H-silt, H-clay) and non-hydrolyzable C fraction (NH-silt, NH-clay), respectively.

All solid fractions were oven-dried at 60 °C prior to weighing. C content and its $\delta^{13}$C in the solid fractions were measured on a thermal combustion elemental analyzer (Fisher Flash 2000, Thermo Fisher, USA) interfaced with a stable isotope Mass Spectrometer (Delta V Advantage, Thermo Finigan, Germany). C content in the water solution was determined by TOC (Vario TOC, Elemental, Germany).

The $\delta^{13}$C of WEOC and DOC was determined by sodium persulfate oxidation to transform the liquid into gas modified by *Midwood et al. (2006)* and *Garcia-Pausas & Paterson (2011)*. Briefly, 10 mL water solution was added into a 250 mL reaction bottle and then 100 $\mu$L of 1.3 mol L$^{-1}$ phosphoric acid solution was added to remove inorganic C from the solution. After that, 200 $\mu$L of 1.05 mol L$^{-1}$ sodium persulfate was added. In the blank control, 10 mL ultrapure water was added, and the other steps were the same. The reaction bottle was then capped and flushed with $CO_2$-free air. The sample was then heated in a 90 °C water bath for 30 min to promote the oxidation reaction. The gas in the bottle was transferred into airbag to determine the $CO_2$ isotope value by Carbon isotope analyzer (912-0003, LGR, USA).

The seven fractions were further divided into three functional C pools based on the supposed relationship between the soil fractions and the stabilization mechanisms: non-protected C pool, chemically protected C pool and biochemically protected C pool modified from *Six et al. (2002)*. The constitute of three functional C pools was shown in Fig. S1.

## Calculations

The soil $CO_2$ efflux rate derived from native SOC and glucose-C during incubation was calculated in *Tian et al. (2016)*.
In the incubation experiment, the amount of primed C and relative magnitude of PE was calculated as follows:

$$AR_{primed} = AR_{SOC(treatment)} - AR_{SOC(control)} \tag{1}$$

$$Relative\ PE\ (\%) = AR_{primed}/AR_{SOC(control)} \times 100 \tag{2}$$

where $AR_{primed}$ is the amount of primed C during the incubation. $AR_{SOC(treatment)}$ is the soil cumulative released-C derived from SOC in the treatment with glucose addition, $AR_{SOC(control)}$ is cumulative released-C in control.

The percentage ($f$) of glucose-derived C in bulk and soil fractions was calculated according to simple isotopic mixing model:

$$f = (\delta^{13}C_{treatment} - \delta^{13}C_{control})/(\delta^{13}C_{Glu} - \delta^{13}C_{control}) \tag{3}$$

where the $\delta^{13}C_{treatment}$ and $\delta^{13}C_{control}$ were the $\delta^{13}C$ value of each C fractions in the treatments with glucose addition and control, respectively. $\delta^{13}C_{Glu}$ is the isotopic signature of the glucose C added to the soil samples. The amounts of new C in bulk and soil fractions were calculated according to C content in bulk soil and soil fractions as well as $f$.

The sequestration potential for glucose-C (proportion of glucose-C retained in soil) was calculated as follows:

$$C\ sequestration\ potential\ (\%) = C_{retained}/C_{addition} \times 100 \tag{4}$$

where $C_{retained}$ is the amount of glucose-C retained (new C) in soil, $C_{addition}$ is the amount of added glucose.

The content of net sequestrated C was calculated as the difference between $C_{retained}$ and $AR_{primed}$. The net C sequestration potential was calculated as the content of net sequestrated C per unit of added glucose-C:

$$Net\ C\ sequestration\ potential\ (\%) = (C_{retained} - AR_{primed})/C_{addition} \times 100 \tag{5}$$

The microbial metabolic quotient ($qCO_2$) was calculated as the $CO_2$ efflux rate per unit of MBC on day 20.

## Statistical analysis

The differences in $CO_2$ efflux rate, isotope signature of each soil fraction, and $qCO_2$ among the three treatments and two soil depths were analyzed using two-way ANOVA. The effects of soil depth and N availability on the amount of primed C, the relative magnitude of PE, the amount of retained glucose C, the sequestration potential, the net C sequestration, and the distribution of new C in each soil fraction were compared using two-way ANOVA. Tukey's post hoc test was used to identify significant differences at $p < 0.05$. The distributions of new C and native C in each soil fraction were compared through independent t test. Principal component analysis (PCA) was used to analyze the distribution of new C and native C in soil fractions. Permutational multivariate analysis of variance (PERMANOVA) with Euclidean distance matrixes was used to evaluate the difference of distribution between new C and native C under the two soil depths. Statistical analyses were performed using SPSS version 21.0. The PCA was conducted using the ''vegan'', ''lattice'', ''permute'' packages in R version 3.5.1. The PERMANOVA analysis was performed using Past 3.

**Table 2** The amount of primed C, the relative PE, the amount of retained glucose C, the C sequestration potential, the amount of net retained C and the net C sequestration potential after 20 days incubation.

| Parameters | Surface soil | | Deep soil | |
|---|---|---|---|---|
| | Glu | Glu+N | Glu | Glu+N |
| Added glucose-C ($\mu$g C g$^{-1}$ soil) | 2061.8 | 2061.8 | 154.7 | 154.7 |
| Primed C ($\mu$g g$^{-1}$) | 257.1 $\pm$ 14.2Aa | 57.7 $\pm$ 9.5Ab | 17.0 $\pm$ 0.3Ba | 8.9 $\pm$ 0.2Bb |
| Relative PE (%) | 28.2 $\pm$ 0.01Aa | 6.3 $\pm$ 0.01Ab | 36.6 $\pm$ 0.0Ba | 19.1 $\pm$ 0.0Bb |
| Retained glucose C ($\mu$g g$^{-1}$) | 872.6 $\pm$ 2.5Aa | 878.0 $\pm$ 2.5Aa | 77.2 $\pm$ 1.7Ba | 79.5 $\pm$ 3.9Ba |
| The C sequestration potential (%) | 43.6 $\pm$ 0.1Aa | 43.9 $\pm$ 0.1Aa | 49.9 $\pm$ 1.1Ba | 51.4 $\pm$ 4.3Aa |
| The amount of net retained C ($\mu$g g$^{-1}$) | 615.5 $\pm$ 14.2Aa | 820.3 $\pm$ 9.5Ab | 60.1 $\pm$ 0.3Ba | 70.6 $\pm$ 0.2Bb |
| Net C sequestration potential (%) | 29.9 $\pm$ 0.7Aa | 39.8 $\pm$ 0.5Ab | 38.9 $\pm$ 0.2Ba | 45.6 $\pm$ 0.1Bb |

**Notes.**

Different capital letters indicate a significant difference between surface soil and deep soil within the same treatment, and different lowercase indicates a significant difference between Glu (single glucose addition) and Glu+N (glucose plus N addition) within the same soil depth. These values are means $\pm$ SE ($n = 3$). There was no glucose addition in the control treatment, the values were 0, hence the control treatment was not listed in the table.

## RESULT

### SOC mineralization and glucose-C retention

Glucose addition increased the $CO_2$ efflux rate (Fig. S2). One part of the increased $CO_2$ efflux was from the added glucose (glucose-derived $CO_2$), while the other part was from the native SOC (native SOC-derived $CO_2$). The amount of primed C and the relative magnitude of PE were significantly affected by soil depth and N availability, and no interaction effect between soil depth and N availability was observed (Table 2 and Fig. S1). The relative magnitude of PE was higher in deep soil than that in surface soil, and N addition significantly decreased them in both soil depths.

After 20 days of incubation, the amount of retained glucose-C and the C sequestration potential were significantly affected by soil depth, but showed no difference between Glu and Glu+N treatments in both soil depths (Table 2). The retained glucose-C was averaged 875.3 $\mu$g g$^{-1}$ and 78.3 $\mu$g g$^{-1}$ in surface soil and deep soil, respectively (Table 2). Deep soil had significantly higher C sequestration potential (49.9 $\pm$ 1.1%) than surface soil (43.6 $\pm$ 0.1%).

The net C sequestration was quantified by calculating the trade-off between the retained glucose-C and the primed C. At the end of 20 days incubation, the retained glucose-C was higher than the amounts of primed C, resulting in overall positive net C sequestration for both soil depths (Table 2). The amount of net sequestrated C and net C sequestration potential was significantly affected by soil depth and N availability, and significant interaction effects were also observed. The amount of net sequestrated C was significantly higher in surface soil than that in deep soil, but deep soil had significantly higher net C sequestration potential than surface soil. N addition could increase the amount of net sequestrated C and net C sequestration potential for both soil depths (Table 2).

### C distribution in soil fractions

Glucose addition significantly increased the isotope signature of bulk soil and the seven fractions (Fig. 1). Compared to Glu treatment, Glu+N increased the isotope signature of

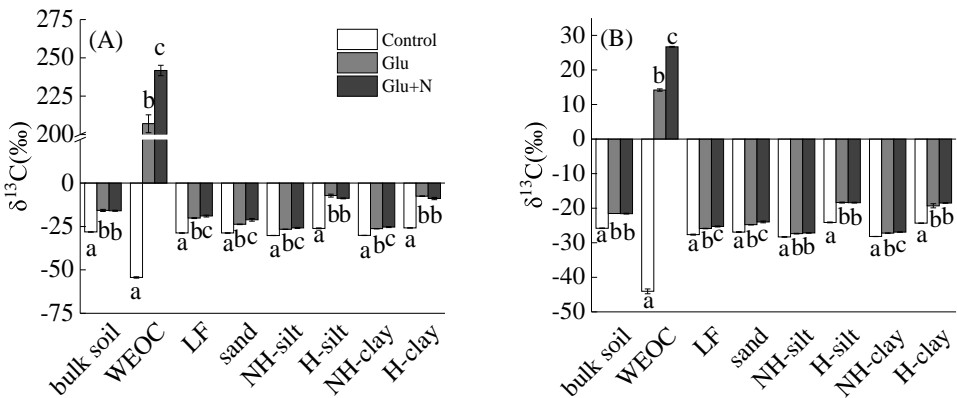

**Figure 1** The isotope signatures of bulk SOC and SOC fractions in surface soil (A) and deep soil (B). Results are means ± SE ($n = 3$). Different letters indicated a significant difference among the three treatments in the same fraction.

**Table 3** The effect of soil depth and N addition on the new C distribution in soil fractions.

| Treatment effect | Soil fractions | | | | | Silt and clay fractions | | | |
|---|---|---|---|---|---|---|---|---|---|
| | WEOC | LF | sand fraction | silt fraction | clay fraction | H-silt | NH-silt | H-clay | NH-clay |
| Depth | *** | *** | * | *** | *** | ns | *** | *** | *** |
| Nitrogen | * | ns | * | ns | ns | ns | * | ns | *** |
| Depth*Nitrogen | ns | ns | ns | ns | ns | ns | ns | ns | ns |

Notes.
\*$p < 0.05$.
\*\*\*$p < 0.001$.
ns, not significant.

LF, sand, WEOC, and NH-clay fractions in both soil depths, but had no significant effect on bulk soil.

The distributions of the new sequestrated C in soil fractions were significantly affected by soil depth and N availability, and interaction effects between soil depth and N availability were not observed (Table 3 and Table S2, Figs. 2 and 3). Compared to the deep soil, the new C in surface soil was more distributed in WEOC (8.5% VS 5.1%), LF fraction (6.9% VS 1.9%) and silt fraction (58.9% VS 43.8%). In contrast, the proportion of new C associated with clay fraction was higher in deep soil than that in surface soil. When silt and clay fractions were further hydrolyzed by acid, the new C associated with silt and clay fractions in deep soil was more acid hydrolyzable (91%) than that in surface soil (80%). N addition slightly increased the proportion of new C incorporated into WEOC, sand fraction and NH-clay fraction for both soil depths, whereas showed no significant effects on other fractions. According to the PERMANOVA analysis, the new C distribution pattern was significantly affected by soil depth ($p = 0.0009$), but not N addition ($p = 0.14$).

The seven SOC fractions were grouped into three functional SOC pools for further analysis (Table 4). Compared to deep soil, the new C in surface soil was more distributed in non-protected C pool (15.8% VS 8.4%) and biochemically protected C pool (15.5% VS
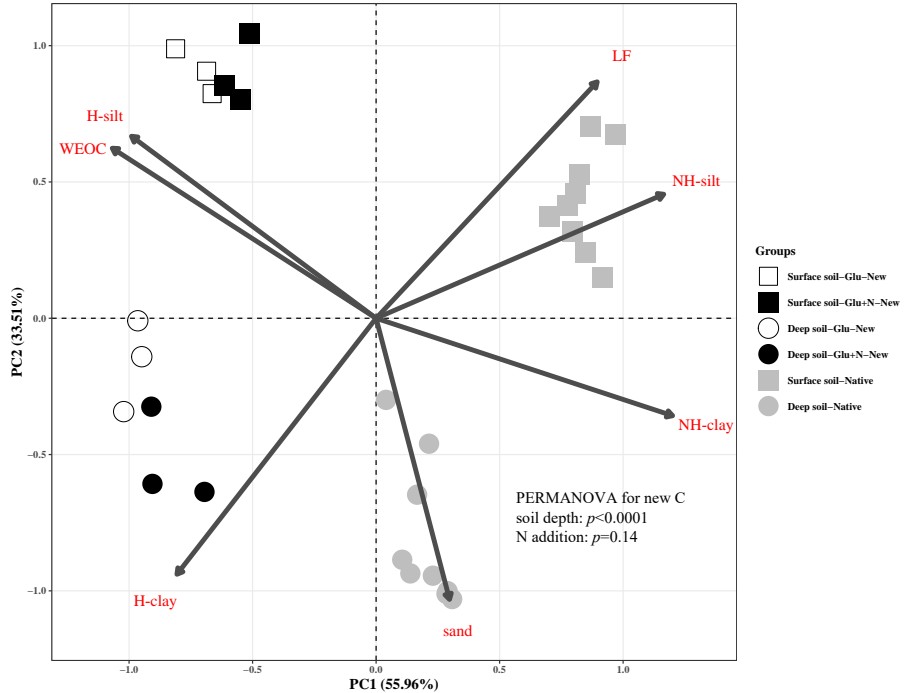

**Figure 2  Principal Component Analysis (PCA) of the distribution patterns of new and native C in surface soil and deep soil.** Square represents the surface soil and circle represents the deep soil. Hollow and black solid denotes new C under Glu and Glu+N treatments, respectively, and gray solid denotes native C. The results of PERMANOVA show the effects of soil depth and N addition on the new C distribution pattern.

7.0%) and less distributed in chemically protected C pool (58.8% VS 74.7%). N addition significantly increased the proportion of new C incorporated into non-protected C pool and biochemically protected C pool in deep soil ($p < 0.05$), and increased proportion of the new C incorporated into biochemically protected C pool ($p < 0.05$) in surface soil.

The distribution of the new C in the seven fractions also differed significantly with native C for both soil depths (Fig. 2). Compared to the native C, the new C was more distributed in non-protected C pool and chemically protected C pool, and less distributed in biochemically protected C pool ($p < 0.05$).

## Soil $qCO_2$

The $qCO_2$ in deep soil was significantly lower than that in surface soil ($p < 0.05$) (Fig. 4). In deep soil, treatments of Glu+N and Glu had significantly lower $qCO_2$ than control treatment. In surface soil, $qCO_2$ showed no significant difference among the three treatments.

**Table 4  The distribution of new C and native C in the three functional SOC pools.**

| SOC pools | Treatments | New C% in bulk new C | | Native C% in bulk C | |
|---|---|---|---|---|---|
| | | Surface soil | Deep soil | Surface soil | Deep soil |
| *Non-protected C pool* | | | | | |
| | Glu | $14.4 \pm 0.3Aa^{a}$ | $7.6 \pm 0.4Ba$ | $11.8 \pm 0.3$ | $6.4 \pm 0.5$ |
| | Glu+N | $17.1 \pm 1.1Aa^{a}$ | $9.1 \pm 0.3Bb$ | | |
| *Chemically protected C pool* | | | | | |
| | Glu | $60.2 \pm 0.4Aa^{a}$ | $75.8 \pm 2.6Ba^{a}$ | $45.1 \pm 0.9$ | $58.1 \pm 1.1$ |
| | Glu+N | $57.3 \pm 0.9Ab^{a}$ | $73.6 \pm 3.0Ba^{a}$ | | |
| *Biochemically protected C pool* | | | | | |
| | Glu | $14.1 \pm 0.1Aa^{a}$ | $6.1 \pm 0.5Ba^{a}$ | $51.9 \pm 0.7$ | $27.3 \pm 0.7$ |
| | Glu+N | $16.9 \pm 1.0Ab^{a}$ | $7.8 \pm 0.3Bb^{a}$ | | |

**Notes.**

For the same functional C pool, different lowercase indicates a significant difference among different treatments within the same soil depth and different capital letters indicate a significant difference for new C in the same treatment between surface soil and deep soil.

[a] Indicated that the incorporation proportion of new C in the fraction was significantly different with the native C. The values for new C are means $\pm$ SE ($n = 3$) and the values for native C are the means $\pm$ SE ($n = 9$) from all treatments.

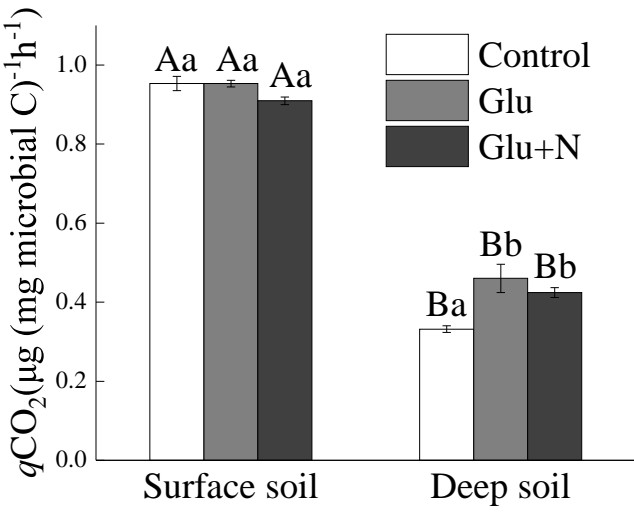

**Figure 4  The effects of glucose and N addition on $q\,CO_2$.** The lowercase letter indicated the difference among the different treatments in the same soil depth and capital letters indicated a significant difference between surface soil and deep soil in the same treatment. These values are means $\pm$ SE ($n = 3$).

## DISCUSSION

### The retention of glucose-C

Labile C input not only altered the native SOC mineralization, but also resulted in new C formation and sequestration (*Cotrufo et al., 2015*; *Haddix, Paul & Cotrufo, 2016*). Previous studies have demonstrated that added glucose can be completely mineralized and assimilated by microbes within 5–7 days (*Coody, Sommers & Nelson, 1986*; *Baldock et al., 1989*; *Lundberg, Ekblad & Nilsso, 2001*; *Zhang et al., 2015*). First, microbes utilized

the glucose for biomass production through the vivo turnover pathway. Second, the glucose-derived microbial residues and their metabolites could be selectively absorbed by soil minerals, and then incorporated into organo-mineral complexes (*Liang, Schimel & Jastrow, 2017*). Furthermore, the contents of glucose-derived DOC (extracted by 0.05 mol $L^{-1}$ $K_2SO_4$ for the fresh soil samples at the end of the 20 days incubation) accounted for 0.07%–0.13% and 0.3%–0.56% of the added glucose in surface soil and deep soil, respectively (Fig. S3), which indicated that little free glucose left in the soil. Thus, we considered that the retained glucose-C mostly existed as live MBC, microbial necromass or microbial metabolites (*Liang, Schimel & Jastrow, 2017*; *Wang et al., 2020*), and soil microorganisms contributed a great role on the C sequestration, especially for the simple C sources (*Wardle, 1992*; *Xu, Thornton & Post, 2013*; *Liang, Schimel & Jastrow, 2017*). Since 80% of the retained glucose-C could incorporate into soil particles eventually (*Griepentrog et al., 2014*), the retained glucose-C in the soil was considered as new sequestrated C at the end of incubation.

Our results suggested that deep soil could retain more proportion of exogenous C than surface soil, which supports our first hypothesis. This could be explained by microbial C use efficiency. $qCO_2$ has been used as a proxy of microbial C use efficiency. The lower $qCO_2$ values in deep soil (Fig. 3) corresponded to higher C allocation to MBC than to respiration losses, indicating a higher C sequestration potential for glucose-C (*Chen et al., 2018*). Previous studies also showed that the $qCO_2$ decreased with soil depth in forest soils (*Spohn & Chodak, 2015*). However, *Spohn et al. (2016)* suggested that CUE varied little with soil depth in two forests. The discrepancy may be explained by the C and N availability or the differences in MBC and community composition (*Spohn & Chodak, 2015*). Additionally, SOC in deep soil was further away from C saturation than that in surface soil. According to the conceptual model of C saturation, the C-poor deep soil would probably have greater potential and efficiency to retain exogenous C (*Stewart et al., 2008*; *Poirier et al., 2013*).

N addition showed no effect on the amount of retained glucose-C, which was contrary to our second hypothesis. It was likely that N addition did not significantly affect the microbial C use efficiency, then did not change the proportion of new C sequestration. This result concurred with the report that N addition did not affect the amount of litter-derived SOC in soil, and suggested that the high N availability in agroecosystem might not affect the C sequestration (*Gentile, Vanlauwe & Six, 2011*).

## The stabilization of new sequestrated C

To better evaluate the stabilization of new sequestrated C, we divided the SOC into different fractions. Labile C addition significantly increased the isotope signature of the soil fractions in both soil depths (Fig. 1), which indicated that glucose-C could be immobilized into all soil fractions. About 80% of the new C was associated with silt and clay minerals for both soil depths. This result was in line with other researches, which found exogenous labile C could be utilized by microorganisms and transformed into mineral-stabilized C as microbial necromass quickly (*Bird, Kleber & Torn, 2008*; *Liang, Schimel & Jastrow, 2017*; *Garten & Wullschleger, 2000*). *Cotrufo et al. (2015)* also suggested that 68% of the litter-derived C in soil was recovered in the mineral-associated silt and clay fraction in the

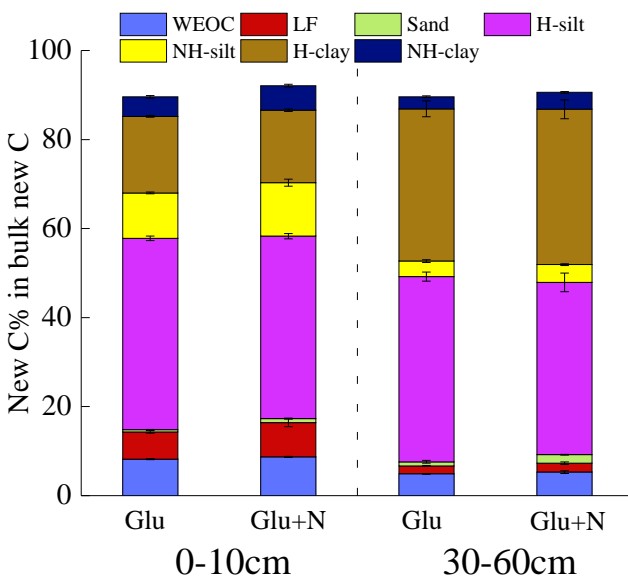

**Figure 3** The distribution of new C in the measured soil fractions after 20 days incubation.

early stage of litter decomposition. Additionally, there was 5–8% of the new sequestrated C distributed in WEOC. The new C in WEOC was thought to be the labile C pool and could be mineralized in the foreseeable future. Overall, associated with soil silt and clay minerals were the main stabilization process for the new C.

Soil depths had a significant effect on the distribution of new C in soil fractions (Fig. 2, Table 3 and Table S2). Compared to surface soil, a higher proportion of new C was stabilized by clay minerals and a lower proportion of new C were stabilized by silt minerals in the deep soil. This could be explained by a higher proportion of clay particles and a lower proportion of silt particles in the deep soil. New C in surface soil was more distributed in NH-silt (11.1% VS 3.7%) and NH-clay (4.9% VS 3.3%) than that in deep soil, which may be attributed to the different microbial products and mineralogy composition in different soil depths (*Kögel-Knabner, 2002*; *Silveira et al., 2008*; *Kallenbach, Frey & Grandy, 2016*). Thus, the distribution of new C might partly depend on the soil properties.

Although N addition had no effect on the total amount of glucose-C retained in soil, N addition could slightly increase the incorporation of glucose-C into WEOC and sand fraction. The higher distribution of glucose-C in WEOC and coarse fractions under N addition were also reported by *Griepentrog et al. (2014)* and *Hagedorn, Spinnler & Siegwolight (2003)*. The decreased turnover of these non-protected fractions under N addition may be responsible for the results (*Gentile, Vanlauwe & Six, 2011*). N addition could increase the N-rich microbial products which could be preferentially associated with the organo-mineral complexes rather than directly attached to the mineral surface (*Kopittke et al., 2018*). This could explain the higher non-hydrolyzable fraction in clay fractions under N addition in both soil depths.

In this study, we divided the SOC into three functional C pools, namely non-protected C pool, chemically protected C pool and biochemically protected C pool. The non-protected C pool was not stable and could be easily mineralized (*Von Lützow et al., 2007*; *Battin et al., 2009*; *Kindler et al., 2011*). Chemically protected C (H-silt and H-clay) is protected by association with mineral particles, where biochemically protected C is a non-hydrolyzable fraction that is stabilized by its inherent complex biochemical resistance through condensation and complexation reactions (*Six et al., 2002*). Nearly 70% of the new C distributed in chemically protected pool revealed that chemical protection contributes most to C stabilization. Compared to surface soil, more new C distributed in clay fraction and less new C distributed in non-protected C pool for deep soil suggested that new C in deep soil was more stable than that in surface soil, which supports our first hypothesis. This result was consistent with previous studies that SOC in deep soil was more stable due to its microbial origin and intimate association with minerals (*Rumpel & Kögel-Knabner, 2011*). Although N addition increased the distribution of new C in non-protected C pool and biochemically protected C pool, PERMANOVA analysis indicated N addition had little effect on the overall new C distribution pattern (Fig. 2). This result suggested that the effect of N addition on the new C stability was limited in the short term.

Compared to the native C for both soil depths, the new C was distributed more in non-protected C pool and less in biochemically protected C pool, indicating that the new C may be less stable than native C. This result was consistent with previous studies that the new incorporated C were more decomposable than native SOC (*Derrien et al., 2014*; *Van Groenigen et al., 2017*). Solid-state nuclear magnetic resonance (NMR) spectroscopy analysis also showed that the glucose-C was transformed mostly into O-alkyl C with little into aromatic C (*Baldock et al., 1989*). Low stability of new C in soil may overestimate the potential of exogenous C sequestration in the short-term. It's notable that we here added the simple decomposable glucose into soil, the proportion of exogenous C retained and its incorporation into SOC fractions might be different from the complex substrates (e.g., litter and root).

## The balance between primed C and new C

We observed positive PE and new C sequestration in both soil depths. The effects of labile C input on soil C pool should be evaluated in the context of net C sequestration (*Qiao et al., 2014*). In the present study, both soil depths showed the positive net C sequestration indicated by the higher amount of new sequestrated C than the primed C in bulk soil. The net increase in SOC was about 35% of the added labile C which was very similar to the meta-analysis results in *Liang et al. (2018)*. Although deep soil had a relatively higher PE, the net C sequestration was higher than that in surface soil. In contrast with our results, previous studies showed that SOC pool in deep soil did not increase under the increased exogenous C input (*Mobley et al., 2015*). The reasons for these divergent results might be that the quantity and quality of the exogenous C and soil type were different. Recent studies had demonstrated that when exogenous C was low, the SOC replenishment from exogenous C could not compensate for the loss of native SOC (*Xu et al., 2019*). Notably, the amount of added glucose-C in this study was different between the two soils (corresponding 100%

of soil MBC). When the same amount of glucose was added, the pattern of C sequestration potential between surface soil and deep soil would change and depend on the amounts of added substrates. Thus, the wide application of these results should be cautious. Future studies with more soil types are needed to investigate the effect of quantity and quality of the exogenous C on net C balance.

N addition could enhance positive net C sequestration through decreasing SOC mineralization (Table 2). The declined PE under N addition could be explained by "microbial nitrogen mining" hypothesis which assumes that an increase in N availability will reduce microbial activity in mining SOC (N-containing substrates in soil) to meet N requirement (*Fang et al., 2018*; *Chang et al., 2019*). The significant interaction effect of N and soil depth suggested that the surface soil tended to retain more net C under N addition. This result demonstrated that future N deposition may favor soil C sequestration by reducing recalcitrant SOC degradation.

Our study suggested that deep soil had higher C sequestration potential than surface soil and N addition could improve the net C sequestration, but widely extrapolating of these results should be cautious. First, all soil samples were sieved through a 2 mm sieve, which might liberate some physical protected SOC and increase their accessibility to the microbes. The changed soil environments might obscure the results. Second, the 20-day incubation could not represent the long-term soil C cycling process. Long-term experiments were needed to monitor the processes of C sequestration. Third, new C sequestration also varied with the quality and quantity of the exogenous C. Four, compared to N availability, P availability was suggested to be more important to soil C cycling in tropical and subtropical forests (*Hui et al., 2019*). The ignoring of P availability and CNP stoichiometry in this study would constrain the application of our results. Extending studies considering these multiple factors could improve our understanding on C sequestration potential for subtropical forest soils.

## CONCLUSION

In summary, labile C addition could result in positive PE in both soil depths, leading to a loss of native SOC. Additionally, labile C input could be sequestrated in soil and overcompensated the C loss induced by PE. Deep soil could sequestrate more proportion of added glucose-C than surface soil, resulting in greater net C sequestration. N addition further increased the positive net C sequestration by decreasing native C mineralization rather than through increasing glucose-C retention. The C distribution in soil fractions suggested that the new C in deep soil was more stable than that in surface soil, and the new C was less stable than the native SOC. Deep soil could retain more proportion of exogenous C with higher stability, suggesting that the deep soil could play a greater role on the C sequestration and stabilization. The lower stability of the new C suggested the soil sequestration potential for exogenous C could be overestimated in short term studies. Future studies with long-term lab incubations and field studies are needed to explore the controlling factors that mediate net C balance and new C stability.

### Funding

This research was supported by the Natural Science Foundation of China (31870465, 31600377, 31700462). The funders had no role in study design, data collection and analysis, decision to publish, or preparation of the manuscript.

### Grant Disclosures

The following grant information was disclosed by the authors:
Natural Science Foundation of China: 31870465, 31600377, 31700462.

### Competing Interests

The authors declare there are no competing interests.

### Author contributions

- Chang Liao conceived and designed the experiments, performed the experiments, analyzed the data, prepared figures and/or tables, authored or reviewed drafts of the paper, and approved the final draft.
- Dong Li performed the experiments, prepared figures and/or tables, and approved the final draft.
- Lin Huang performed the experiments, authored or reviewed drafts of the paper, and approved the final draft.
- Pengyun Yue analyzed the data, prepared figures and/or tables, and approved the final draft.
- Feng Liu conceived and designed the experiments, authored or reviewed drafts of the paper, and approved the final draft.
- Qiuxiang Tian conceived and designed the experiments, analyzed the data, prepared figures and/or tables, authored or reviewed drafts of the paper, and approved the final draft.

### Field Study Permissions

The following information was supplied relating to field study approvals (i.e., approving body and any reference numbers):

Field sampling was verbally permitted by Zhirong Gu, who is a worker at the Badagongshan National Research Reserve.

### Data Availability

The raw measurements are available in a Supplemental File.

### Supplemental Information

Supplemental information for this article can be found online at http://dx.doi.org/10.7717/peerj.9128#supplemental-information.

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
