# Peer review of "Higher carbon sequestration potential and stability for deep soil compared to surface soil regardless of nitrogen addition in a subtropical forest"

_PeerJ, doi:10.7717/peerj.9128_

## Round 0.1 · original submission · Minor Revisions

Your work has been reviewed by three experts. I agree with their comments. Please consider them in revision.

Reviewer 1 ·

Basic reporting

The authors provided adequate background information related to this study. The manuscript was well organized. Tables and figures are nicely presented. Hypotheses could be improved by providing explanation or reasons for the two hypotheses. English writing is fine in general. But there are a few errors and awkward sentences that need to be corrected.

Experimental design

The research is original and fit the aims and scope of the journal. Rational was provided in the introduction. Materials and methods were provided in details. For data analysis, I think the authors could test glucose addition, N addition and their interactions. Thus, a three-factor ANOVA should be used. It seems that the authors only tested the simple treatment effects (compared among CK, Glucose, N, and Glu+N treatments), plus depth.

Validity of the findings

The authors clearly provided the results, and results are generally sound. The conclusion was supported by the results.

Additional comments

Overall, this is a well-designed lab incubation experiment. Although the incubation was short, the trends of CO2 emissions were clear. Results were well presented and explained in the discussion. I don’t have any major concerns with this study, but I have two suggestions for the authors. One is the data analysis. The authors used a two-factor experimental design with N addition and glucose addition. But ANOVA only considered simple effects of four treatments. Glucose, N and their interaction could be test. Another one is that I feel that the results and conclusion of “subsoil has a greater potential in long-term C sequestration in forest ecosystems” could be caused by the experimental design. Different amounts of glucose were added to topsoil and subsoil. If the same amount of glucose was added, the results could be different. I think the authors need to address this issue, at least to discuss this point. I recommend minor revision.
Specific comments;
L16: change “(priming effect)” to “through priming effect”?
L75: change “widely founded” to “mostly found”
L87-89: Please add explanations to the two hypotheses.
L101: topsoil and subsoil: Usually subsoil is just below the topsoil. As 10-30 is transition soil layer, why did not include this layer as well?
L220: delete by
L223: t test
L288: I recommend a reference showed similar phenomena (Hui et al. 2019. Phosphorus rather than nitrogen enhances CO2 emissions in tropical forest soils: Evidence from a laboratory incubation study, EJSS).
L292-293: I don’t agree with this statement. What would happen if the same amount of glucose was added to the subsoil?
Table 2. Please add the CK to the table.

Reviewer 2 ·

Basic reporting

no comment

Experimental design

no comment

Validity of the findings

no comment

Additional comments

This manuscript is well and informative. However, there are still some problems that need to be revised before publication. The manuscript has two major weaknesses. First, the distribution of 13C-glucose into microorganism’s pool is lack of direct data supporting. If the author could strengthen the discussion by adding some microbial utilization mechanism will greatly improve the manuscript. Second, using the results of a short-term laboratory incubation to explore long-term phenomenon of fresh C partitioning and stabilization, thus, the results should be interpreted with caution.


L119-120 The incubation in dark? Oxygen condition? And how to control soil moisture during incubation?
L134 Why not use the same jars for CO2 efflux rate and 13C-CO2 determination?
L147 I think the Fig. 1 belong to the appendix.
L236 mg g-1 should be “ug g-1”
L242 “affect the C” delete extra spaces
L289 “retained glucose-C mostly existed as live MBC…” Can you show the range of data for mostly? Is there really no free glucose left?
L294 Same substrate (glucose) lead to different microbial CUE of Top- and Sub- soil. This should be further explained. Microbial community compositions differ in the two depth soils? Or soil characteristics difference result in different CUE?
L306-308 I very agree with you. Long-term studies are needed. However, the limitation of the current study should be pointed out.
L321-322 “the 13C signature in WEOC unlikely comes from free glucose, but from the live MBC” need more supporting evidence.
L369 4.2 to 4.3
L397 priming effect to PE
L402-403 “suggesting that the subsoil played a greater role in the long-term C sequestration and stabilization” this conclusion should be caution due to the only 20 days lab incubation.
Figure 2, the Y-axis scale is not appropriate for A.
Table 1 WEOC data?
Table 2 The amount… unit
Table 3 I recommend the pie figure for these data.
Figure S1 The cumulative CO2 release for 13C-glucose also should be given.

Reviewer 3 ·

Basic reporting

The MS "Higher carbon sequestration potential and stability for forest subsoil compared to topsoil regardless of nitrogen addition in a subtropical forest" describes an interesting work to elucidate the carbon sequestration potential of surface and subsoil in a tropical forest and how nitrogen addition could affect this carbon sequestration. Laboratory incubation experiments were performed for 20 days and 13C-labelled glucose was used as C substrate. MS is based on sound literature and background information, description of results seem adequate and practical, and discussion section is also relevant. The authors have critically discussed their hypotheses. I have provided my feedback directly to the pdf copy of the MS to help the authors during revision process

Experimental design

Experimental design is OK as are the statistical analysis. Experimental design is also practical in lieu with the hypotheses. As mentioned as comments/suggestions, I was only curious about authors not using 13C-labelled litter in their experiments.

Validity of the findings

The study provides important insights into the mechanistic understanding of potential C sequestration in surface and subsoil. The authors found non-significant effects of nitrogen on priming effect and C sequestration. Despite the advocacy of the authors, I think a cautious approach is required while applying the study interpretations at field scale and also on long-term basis. Therefore, the term "C retention" rather "C sequestration" may be more appropriate the context of the study. Level of speculation could be also be used with care

Additional comments

The MS is generally well-written and results and discussion sections have strong implications. However, at points during discussion, the authors have overlooked the role of CNP stoichiometry in C sequestration.

Annotated reviews are not available for download in order to protect the identity of reviewers who chose to remain anonymous.

---

## Round 0.2 · accepted · Accept

Thank you for your careful revision of the manuscript. All three reviewers are satisfied with your revision, and I agree.

Reviewer 1 ·

Basic reporting

This is a revised manuscript. The authors have made efforts and adequately addressed my questions. I don't have any further comments.

Experimental design

It would be better to have a full factor design. But the authors explained why no N alone was applied.

Validity of the findings

Results were well presented and explained.

Additional comments

I recommend Accept.

Reviewer 2 ·

Basic reporting

no comment

Experimental design

no comment

Validity of the findings

no comment

Additional comments

I’m satisfied with the author’s revision. This new version looks good.

Reviewer 3 ·

Basic reporting

N/A

Experimental design

N/A

Validity of the findings

N/A

Additional comments

The authors have responded positively to the comments of the reviewers and manuscript has been revised appropriately. Overall structure and message of the manuscript has also become clear and improved. .